# PYLIC: LEVERAGING SOURCE CODE FOR PLANNING IN STRUCTURED ENVIRONMENTS

## ABSTRACT

This paper investigates the application of program analysis techniques to planning problems in dynamic environments with discontinuities in long-horizon settings. Traditional approaches rely on specialized representations, which are often tailored to specific problems and domains. In contrast, we propose describing the combined planning and control problem directly as a desired property of the execution of simulator source code. This representation is expressive, naturally providing a means to describe desired properties of even very dynamic and discontinuous environments. We show that, despite this generality, it is still possible to leverage domain knowledge by relating it to the simulator source code. We measure the effectiveness of this approach in several case studies in simulated robotic environments. Our results show that in these environments, our framework can improve the efficiency in solving the control and planning problem, relative to standard numerical search and reinforcement learning methods.

## 1 INTRODUCTION

This work is motivated by the challenges present in decision-making in dynamic and highly discontinuous environments over prolonged periods of time. These challenges include discontinuities and non-convexity, rendering the naive application of black-box optimization techniques like gradient descent unsuitable for planning in long-horizon settings.

A standard approach to tackling these challenges is to factorize the problem into discrete task planning through symbolic reasoning and continuous motion planning (Kaelbling & Lozano-Perez, 2011; Fainekos et al., 2009; Plaku & Karaman, 2016; Pinneri et al., 2021; Kim et al., 2017; Dantam et al., 2016; He et al., 2015), allowing domain experts to encode the structured nature of the search space into a symbolic planning domain. This, however, requires the relationship between symbolic plans and low-level dynamics into which the plans can be grounded to be made explicit through an ad-hoc coordination layer. Recent work (Toussaint, 2015; Takano et al., 2021; Leung et al., 2021; Li et al., 2021a; Xiong et al., 2022) leverages logical specifications –which have a long history with software and robotics (Fainekos et al., 2009; Kloetzer & Belta, 2007; Kress-Gazit et al., 2009; Plaku & Karaman, 2016; Maler et al., 2006; Li et al.; Brafman et al., 2018; Giacomo et al., 2019)– to address some of these limitations by directly relating logic semantics to low-level dynamics. This removes the need for an ad-hoc layer between the symbolic and low-level planners, and allows scalable numerical optimization techniques to be applied. These approaches, however, deliberately ignore the structure in the simulator –e.g., syntactic features like control flow statements– when describing specifications, instead treating the simulator as a differentiable black-box.

Model structure has been established as a powerful source of information to tackle the challenges present in non-convex and discontinuous settings. One approach to leverage structure is to smooth discontinuities (Chaudhuri & Solar-Lezama, 2011; Pang et al., 2023; Duchi et al., 2012; Posa et al., 2014; Howell et al., 2022), which can mitigate some problems in applying gradient-based search to discontinuous systems. Smoothing, however, fundamentally relies on hiding discontinuities, even though they may be useful in deriving a solution (Bangaru et al., 2021).

In contrast, we approach the challenges present in dynamic environments by explicitly relating domain knowledge to runtime information of the simulator, as well as to its syntactic structure. Our proposed methodology applies insights from techniques in the field of dynamic program analysis, such as concolic testing, as well as from planning approaches involving symbolic search and logical

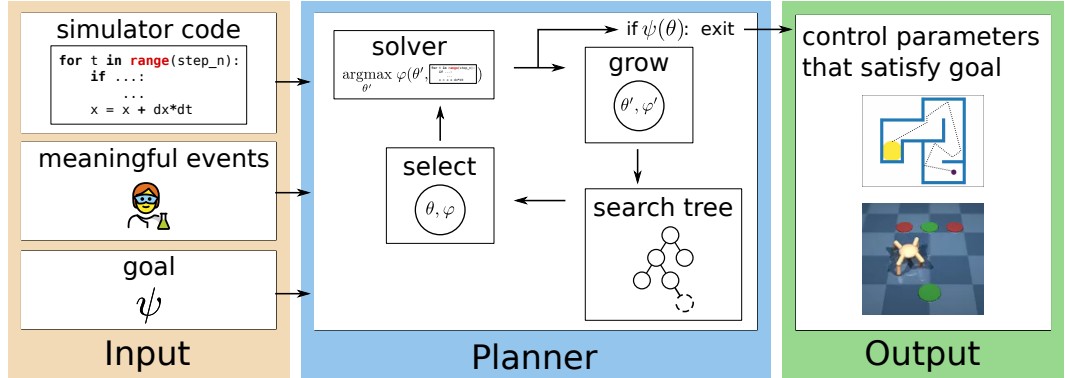

Figure 1: Overview of the proposed framework. The user provides simulator source code, a goal, and "meaningful events" –high-level properties useful for solving the task, expressed as properties of execution of the source code–. The framework then generates a planner that constructs a search tree to guide the search for parameters that satisfy the goal. Numerical search is used to find parameters that satisfy predicates in the search tree. The planner exits upon finding control parameters that satisfy the goal.

specifications. The key observation is that many of the properties traditionally encoded into ad-hoc planning domains can instead be directly related to simulator source code. This removes the need to create a separate ad-hoc planning domain from scratch because –unlike existing logic-based approaches to robotics– it allows plan descriptions to explicitly leverage and reuse the structure and logic already present in the source code (e.g., "make the condition of this *if statement* true").

Our contribution is thus a framework where the combined planning and control problem is described as a property of the execution of the simulator itself. In our framework, domain knowledge is related to the structure of the simulator source code and used to generate a tree-based planner. This is akin to existing hierarchical planning approaches, except that the control problems in the search tree leverage the structure present in the simulator source code. The control problems can then be solved with scalable numerical search techniques like gradient descent. Figure 1 shows an overview of our approach, discussed in detail in Section 3. We instantiate our framework in a Python-based implementation, and perform case studies on different tasks and simulators, comparing our approach with numerical search and reinforcement learning techniques.

## 2 MOTIVATING EXAMPLE

Consider a two-dimensional continuous dynamical system where the goal is to control a circular body ("marble") from a fixed initial state, $s_0$, to a fixed target position ("goal"), as displayed in Figure 2. At each timestep $t$, a force is applied to the marble. Each force is a two-dimensional vector with entries between -1 and 1, and these numbers correspond to our control parameters $\theta[t]$. The bounds are small enough to make the system under-actuated: i.e. at the speed it must travel to reach the goal, there is not enough force for the marble to make tight turns, and instead it must use the walls to bounce its way to the destination. The marble is affected by drag and by the obstacles (walls) in the maze which the marble can collide with. Thus, each task consists of finding a sequence of thrusts that will take the marble from an initial position to a goal position. A programmatic description of this system is shown in Listing 1.

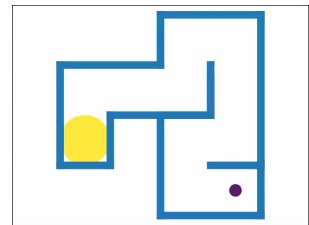

Figure 2: The task consists of applying thrusts to the marble (purple) to reach the target location (yellow), navigating a maze containing obstacles (blue).

A naive approach to solving this problem through numerical optimization is to define a differentiable cost of any particular candidate solution and perform gradient descent. For example, we could use

the distance to the goal after simulating the system as the cost function, and use program smoothing to account for discontinuities. However, as is often the case in discontinuous environments, the resulting optimization problem is non-convex, making gradient-based search inadequate. Similarly, it is challenging to scale symbolic approaches (such as Satisfiability Modulo Theory-based solvers) to highly discontinuous rigid body simulations such as the one under consideration.

Listing 1: Marble simulation code.

```
def simulation(theta: Tensor) -> State:
  state = s0  # initial state
  for t in range(len(theta)):
    # Update state assuming no collisions
    action = theta[t].clip(-1, 1)
    new_state = step_nocoll(state, action)

    # Check for collisions
    for i in range(len(state.obstacles)):
      v = get_distance(new_state, obs_i)
      if v < 0:   # ID: collision_check
        # If collided, adjust next state
        new_state = step_coll(new_state, i)

    # Update state
    state = new_state
  return state
```

Note that, in this case, discontinuities in the dynamics are described by *if statements*. If we constrain the search space to control parameters that induce some particular execution path, then the optimization problem is continuous, and numerical search techniques like gradient descent are likely to find a locally optimal solution. Thus, under mild assumptions, we can (in principle) perform a local search on every execution path until we find a path that leads to a solution to the control problem. In this way, the combined planning and control problem becomes a problem of path-finding inside the simulator. However, naively searching every execution path is unfeasible[1].

Instead, our system avoids the exhaustive search over execution paths by leveraging domain knowledge. For example, a domain expert would know that solutions to the tasks likely induce sequences of "meaningful events" that have a specific structure: "bounce off some unknown sequence of obstacles and then reach the goal". Even though the expert does not indicate a specific sequence of obstacles, constraining the search to executions that induce sequences of meaningful events with that structure aggressively prunes the execution paths that have to be considered. The expert provides this domain knowledge by describing how to construct sequences of meaningful events. The system then incrementally searches over sequences of meaningful events, finding inputs that induce execution paths that match candidate sequences, until a solution to a given task is found.

Meaningful events along a sequence are predicates over the execution of the simulator source code. For example, the meaningful event "collide with obstacle 3 between timesteps 0 and 60" is a predicate that is true only for execution traces in which the condition in the control-flow statement labeled with # ID: collision_check was satisfied when the corresponding variables have appropriate values. Note that this reuse of the simulator code allows the expert to include (e.g.,) collisions in sequences of meaningful events without re-implementing the corresponding logic from scratch.

In the following section, we describe the notation used to describe meaningful events. Importantly, we expect expert users to provide a simulator and describe the meaningful events. Therefore, the notation must be appropriate for use by programmers. We also describe the algorithm used to perform the search over sequences of meaningful events.

## 3 METHODS

We now ground intuitions from the previous section into a planning framework that relates domain knowledge to simulator source code.

**Problem statement**  We deal with continuous sequential decision making problems over a finite horizon of length $T$ where the goal is to find a sequence of time-indexed actions $\theta[t] \in \mathbb{R}^{T \times n}$ that, from a fixed initial state, achieve a goal represented as a predicate, say $\psi(\theta)$, where $n$ is the dimensionality of the actions. Crucially, we also assume we are provided with a simulator of the environment that takes a sequence of actions as input, and that the task has an underlying discrete structure (e.g., low-level discontinuities in the dynamics, or high-level task structure) in which domain experts can identify meaningful events and relate them to the simulator source code, as described below.

---

[1]In this program the number of execution paths is $O(2^{TN})$, where $T$ and $N$ are the number of timesteps and obstacles. Even for small values, say $T = 200$ and $N = 8$, exhaustive search is unfeasible.

| Name | Type | Meaning |
|------|------|---------|
| *input* | $T$ | Constant that holds the complete execution trace. |
| $\neg$ | $B \rightarrow B$ | Standard unary boolean operator. |
| $\wedge, \vee$ | $B \times B \rightarrow B$ | Standard binary boolean operators. |
| if_or | $T \rightarrow B$ | True iff there is a true *IfNode* in $T$. |
| filter | $T \times F \rightarrow T$ | Filter a trace with a filter predicate. |
| $<, >$ | $\mathbb{R} \times \mathbb{R} \rightarrow B$ | Standard inequality predicates. |
| $+, -, *, /$ | $\mathbb{R} \times \mathbb{R} \rightarrow \mathbb{R}$ | Standard arithmetic operators |
| $\mu$ | $T \rightarrow \mathbb{R}$ | User-defined functions which map an execution trace to a real value. |
| $F$ | $T \times \mathbb{N} \rightarrow B$ | User-defined filter predicates which determine if the $i$-th node of a trace belongs to a subset. |
| $c$ | $\mathbb{R}$ | Real-valued numbers. |

Table 1: Trace predicate grammar: any expression whose type is $B$ is said to be a trace predicate. Here, $T$ is the type of execution traces, and $B$ is the set of boolean values.

**Meaningful events**  Often, domain experts know of specific high-level properties of candidate solutions that might be useful for solving a task, and can describe these properties as precise statements about the execution of the simulator over a finite horizon. We refer to these insights as "meaningful events". Note that analogous concepts exist in other planning frameworks (e.g., Toussaint (2015); Shah & Srivastava (2022); Garrett et al. (2020); Hoffmann et al. (2011)). In our motivating example, collisions of the marble with obstacles correspond to meaningful events described as statements about the execution of a specific control-flow condition.

Our method's key assumption is that a successful plan will involve a sequence of meaningful events. This allows the problem of solving for parameters that reach the goal to be decomposed into a series of sub-problems that seek to match the next meaningful event given parameters that successfully matched the previous ones. For example, instead of planning directly for the marble to reach the goal, the system may plan to hit obstacle 1 then, if successful, search for parameters to hit obstacle 2 given the plan to hit obstacle 1, and finally for the parameters which will reach the goal from obstacle 2. Note that the exact meaningful events and the ordering (e.g., which obstacles to collide with and in which order) that will lead to a solution is unknown. Section 3.1 describes the search for a satisfying sequence of meaningful events. We detail the numerical search for parameters that match a given meaningful event in Section 3.2.

**Execution traces.**  Our system instruments programs so that whenever a control flow instruction is executed, a record is created. Records are data structures which contain the program state at recording time and the *ID* with which a control flow structure was labeled, as in Listing 1. We define a trace to be a list of such records from a particular execution (see Appendix A). The tracing system is as a function $\mathrm{tr}(S, \theta)$ which returns the execution trace of a program $S$ on input $\theta$. The system automatically instruments the input source code for tracing (see Appendix C).

**Trace predicates.**  Meaningful events are described in the form of "trace predicates". Trace predicates describe a set of execution paths of interest as a predicate over execution traces of the simulator. The language to describe trace predicates consists of standard boolean logic with arithmetic operators and three constructs specific to the program traces: (1) if_or is a disjunction over the truth values of all if statement records in a trace, (2) $\mu$ are real-valued functions provided by the user which extract run-time values from a given execution trace (e.g., "distance to goal at the end of the trace"), (3) filter describes and filters a subset of a trace, as often only a subset of the trace will be relevant for an event (e.g., "only the collision control-flow condition when $i = 3$"). The result is Table 1. Note that predicates in this language can leverage both the syntactic structure and run-time information of the source code, which is a key property that allows source code to be reused when describing meaningful event sequences. See Listing 2 for an example trace predicate in this language.

Note that, even though one might consider other formalisms to describe trace predicates –e.g., extending this language with temporal operators–, in our experiments we found this language both flexible and easy to use.

## 3.1 SEARCHING FOR SEQUENCES OF MEANINGFUL EVENTS

Listing 2: Trace predicate for colliding when $i = 3$ and $0 \leq t \leq 60$ (see Listing 1).

```
IfOr ( Filter (
  input ,
  lambda ( trace , i ): (
    trace [ i ]. id == " collision_check "
    and 0 <= trace [ i ]. prog_state [ " t " ] <= 60
    and trace [ i ]. prog_state [ " i " ] == 3
 )))
```

The system searches for a sequence of meaningful events that solves a given task. The assumption is that there are sequences of meaningful events that solve a task with the additional property that, given the parameters that satisfy a prefix of the sequence as a starting point, local search can find the parameters that match the next meaningful event in the sequence. Since the exact sequence that satisfies the goal is unknown, the system performs a tree search – a standard approach in many planning frameworks–, where paths in the tree correspond to sequences of meaningful events.

Each node in the tree describes a search problem corresponding to finding control parameters that match a meaningful event given some control parameters so far, with the meaningful event described as a trace predicate.

The routine SOLVE solves instances of such search problems (see subsection 3.2). The user provides routines for choosing a node and for growing a node with children in case the problem in the node is solved, denoted with CHOOSE and GROW respectively. These routines are used by the system to grow the search tree (see Algorithm 1).

---

**Algorithm 1** Planning with execution traces

---
**procedure** TRACEPLANNER($S$, CHOOSE, GROW, $\theta$)
    $D \leftarrow []$                                                   $\triangleright$ *Init. empty dataset*
    $T \leftarrow (\top, \theta)$                                            $\triangleright$ *Init. search tree*
    **while** $\neg$ ISSOLUTION($\theta$) **do**
        $n \leftarrow$ CHOOSE($T, D$)                             $\triangleright$ *Choose node from tree*
        $\theta \leftarrow$ SOLVE($n.\varphi, S, n.\theta$)
        **if** $\rho(\varphi, \text{tr}(S, \theta)) > 0$ **then**               $\triangleright$ *If solver succeeded*
            record $(n, \theta)$ as successful in $D$
            **for** $c = (\varphi', \theta) \in$ GROW($n, T, \theta$) **do**
                add child $c$ to node $n$ in $T$
        **else**
            record $n$ as failed in $D$
    **return** $\theta$

---

## 3.2 FINDING INPUTS THAT MATCH MEANINGFUL EVENTS

Finding input parameters to a program that match a given meaningful event described as a trace predicate corresponds to solving a satisfiability problem. In the context of Signal Temporal Logic, previous work (Takano et al., 2021; Leung et al., 2021; Li et al., 2021a; Xiong et al., 2022) has shown that *quantitative semantics* can be used to leverage numerical search algorithms to solve the satisfiability problem for formulas in continuous systems. Quantitative semantics describe the degree to which a formula is satisfied by defining a *robustness value*, which is a real-valued relaxation of traditional boolean semantics, with positive robustness values indicating satisfaction. We adapt quantitative semantics to trace predicates and denote the robustness value of a predicate $\varphi$ given execution trace $\tau$ as $\rho(\varphi, \tau)$. Please refer to Appendix B for a precise description of the quantitative semantics for trace predicates. Our implementation automatically relaxes compatible arithmetic inequalities in conditional control flow into their equivalent quantitative semantic expressions before tracing (see Appendix C for details).

The satisfiability problem for a trace predicate $\varphi$ and program $S$ is thus the following search problem: find $\theta$ s.t. $\rho(\varphi, \text{tr}(S, \theta)) > 0$, which is then formulated as an optimization problem:

$$\operatorname*{argmax}_{\theta} \quad \rho(\varphi, \text{tr}(S, \theta)).$$

If the satisfiability problem has a solution, then it can be found by solving the aforementioned optimization problem. As an example, consider Algorithm 2, which shows how to leverage gradient-based search (Cauchy, 1847) to solve the satisfiability problem[2]. While the optimization problem might still be non-convex or discontinuous, the assumption behind our approach is that local search is sufficient to find solutions to sequences of meaningful events, provided they are solved incrementally, thus avoiding a global search. Thus, users are subject to the limitations of the chosen numerical search algorithm when defining meaningful events.

---

**Algorithm 2** Solve $\varphi$ on program $S$ with initial params. $\theta$

---

  **procedure** SOLVE($\varphi, S, \theta$)
    **while** $\rho(\varphi, \mathrm{tr}(S, \theta)) \leq 0$ **and not** converged **do**
      $\tau \leftarrow \mathrm{tr}(S, \theta)$            ▷ *Trace program execution*
      $\theta \leftarrow \theta + \lambda \nabla \rho(\varphi, \tau)$     ▷ *Gradient ascent robustness*
    **if** $\rho(\varphi, \mathrm{tr}(S, \theta)) \leq 0$ **then**
      **raise** solver failed
    **return** $\theta$

---

**Summary** To make use of our framework, users provide simulator source code, a goal predicate, and routines that characterize a search over meaningful events. These generate a tree-based planner, which finds a sequence of meaningful events that solve a given task. Meaningful events are represented with trace predicates, which are statements about the execution of the source code, and can reference labeled control-flow structures in the source code. The simulator source code is automatically instrumented so that the satisfiability problems induced by meaningful events in the tree can be solved with off-the-shelf numerical search techniques.

# 4 EXPERIMENTAL EVALUATION

We conduct case studies involving systems with discontinuous dynamics and non-convex tasks. The goal is to measure whether leveraging source code with our framework leads to higher planning performance, compared to other forms of encoding domain knowledge. As described in each subsection, each case study consists of calling Algorithm 1 with experiment-specific CHOOSE, and GROW routines. We highlight the source code structures that were reused to describe sequences of meaningful events. For the SOLVE routine, if the simulator is differentiable, we use gradient descent (see Algorithm 2), and otherwise we use CMA-ES (Hansen & Ostermeier, 1996) as implemented by pycma (Hansen et al., 2019). We implement the framework as an open-source Python package we named **Pylic**. We use Python's introspection capabilities to implement the tracing system, which made it straight-forward to leverage off-the-shelf optimization algorithms in SOLVE routines.

Additionally, in each case study, we compare our framework with two **baseline** approaches, which are given substantial domain knowledge. One of the approaches consists of a Model Predictive Controller (MPC) using the Cross Entropy Method (CEM), a standard trajectory optimization approach (Pinneri et al., 2021); we chose the other approach according to the task under consideration. All approaches are given equal computational resources.

We use two metrics to measure performance: "progress" and "success rate". Progress refers to the fraction of a task that has been solved by an algorithm at a particular planning time, with 0% at the initial state and 100% indicating a successfully completed task. Success rate is the fraction of tasks that an algorithm has successfully completed at a particular time.

## 4.1 MARBLE MAZE

The first case study is the marble maze task described in Section 2, which consists of navigating a ball through a maze by applying bounded thrust at each timestep. We analyze the success rate of the algorithms across 25 randomly generated mazes.

---

[2]The choice of numerical search algorithm is ultimately domain-specific.

**Pylic**   We follow the insights described in Section 2. The CHOOSE routine simply finds the first unsolved node in the tree in depth-first order. The GROW routine takes a solved node, checks which obstacles the marble collided with by tracing the simulator, and returns trace predicates corresponding to satisfying the collision control-flow condition for all further obstacles that have not been collided with (see Listing 2 for an example predicate) and reaching the target position. In the SOLVE routine, we use Pytorch (Paszke et al., 2019) to compute gradients.

**Baselines**   We first compare our framework against the Soft Actor-Critic algorithm (SAC) (Haarnoja et al., 2018) as implemented in Stable Baselines3 (Raffin et al., 2021). SAC is a state-of-the-art off-policy RL algorithm. We leverage our domain knowledge to frame the task as a navigation task across a path that leads to the target position, rewarding the agent to move through a sequence of waypoints, which are given. The relative position of the next waypoint is included in the observations. We follow the reward structure of Sartoretti et al. (2019), providing positive reward upon reaching a waypoint. In this task, we are interested in comparing the efficiency of each algorithm in solving a given task, and thus one RL policy is trained from scratch for each task. For the second approach, we use an MPC using the CEM, with the cost of a candidate trajectory defined as the negative cumulative reward.

## 4.2   PASSWORD LOCOMOTION

This case study is a locomotion task in a three-dimensional simulation using the Mujoco physics engine (Todorov et al., 2012). In this task, there is a robot consisting of rigid bodies connected through joints, a target position, and buttons on the ground which are activated when the robot stands on top of them (see Figure 3). The target position is initially surrounded by walls, so directly navigating to it is impossible. To remove the obstacles, the buttons have to be activated in a particular order ("password"), which is not given to our framework, but is provided to the baselines. The actions consist of torque applied to the robot joints. We test on all passwords up to three buttons.

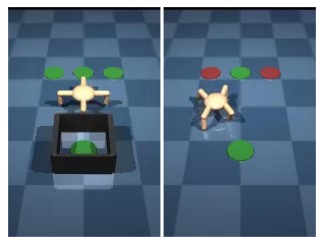

Figure 3: Password locomotion task solved with our framework. In this case, the password is (2, 0).

**Pylic**   In contrast to the previous case study where we used Pylic to leverage low-level dynamics, here we treat the call to the physics engine as a black box, and instead focus on the high-level code structure that checks whether a button is activated, thereby showing that we do not need to reason about simulation code in its entirety to leverage its structure. The CHOOSE routine finds the first unsolved node in the tree in depth-first order. We use CMA-ES for the SOLVE routine. The GROW routine takes a solved node, checks which buttons have been pressed by running the simulator on the node's control parameters and returns a list with trace predicates corresponding to satisfying the button-press control-flow statement for each button that has not been pressed yet, as well as a predicate that encodes reaching the target position.

**Baselines**   We again first compare our framework with the SAC algorithm. We encode the task as navigating through a sequence of waypoints, such that going one after the other solves the goal. A single policy is trained in a multitask fashion across randomly sampled passwords, providing positive reward as the robot moves towards and traverses the waypoints. The observations include the angle positions and velocities of the robot, as well as the relative position of the next checkpoint. We measure whether the policy at a particular training time can solve the tasks given the corresponding sequence of waypoints. Note that a lot of domain knowledge is provided to the baseline, as the waypoints to all tasks are provided (thus, the baseline is always provided the correct password, including at test time). In contrast, our method does not get access to the password, and has to find the password through trial-and-error during the tree search. As before, the second baseline is an MPC using the CEM, with the cost of candidate trajectories being the negative cumulative reward.

## 4.3   MARBLE RUN

In this case study, actions consist of placing platforms over which a marble can slide, with the goal of making the marble collide with all "boxes" in the environment. The marble is dropped from a fixed position above and to the left of the box in the top-left. See Figure 4. Actions consist of four numbers denoting the two-dimensional endpoints that describe a platform. An action can be performed whenever the marble collides with a box. We simulate the low-level dynamics with the Pymunk 2D library (http://www.pymunk.org/), and test across seven manually crafted tasks.

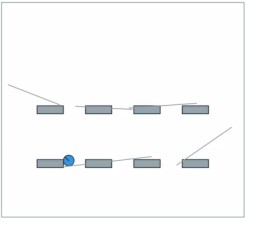

Figure 4: Marble run task solved with our framework.

**Pylic** We treat the low-level calls to the physics engine as a black box and focus on the high-level structure that controls the interaction between box collisions and platform placing. The GROW routine traces the simulator to find the first row with at least one box that the marble did not collide with, and returns a predicate corresponding to satisfying the collision control flow condition for each box in that row that the marble did collide with. We use CMA-ES for the SOLVE routine. For the CHOOSE routine, notice that it is possible to place a platform that causes the marble to collide with the target box, but which makes it impossible to collide with other boxes, so it can be necessary to find multiple solutions for a particular node. Therefore, CHOOSE considers all nodes in the tree and assigns to each a score equal to the number of collisions in that node so far divided by the number of times that the node has been selected before, selecting the node with the highest score.

**Baselines** We first compare with CMA-ES applied to the global problem of maximizing the number of boxes that the marble collided with, optimizing the actions for the entire trajectory at once. This case study thus allows us to compare the benefits of our framework with the corresponding standalone version of the numerical search algorithm in a problem with relatively small dimensionality, a setting in which pure CMA-ES is adequate. Note that CMA-ES is an algorithm noted for its good performance on non-convex and discontinuous optimization problems. If pure CMA-ES converged prematurely without solving a task, we restart the optimization process until the timeout. In our experiment. As in the other case studies, the second baseline is an MPC using the CEM with the same cost function.

### 4.4 RESULTS

Our experimental evaluation shows the effectiveness of the approach, as it results overall in greater success rates than the baselines, which include state-of-the-art RL algorithms and standard numerical search methods (see Figure 5). While the baselines generally make steady progress, they are unable to fully solve many tasks, resulting in relatively low success rates. This demonstrates the difficulty of using numerical search methods –such as CMA-ES, CEM or RL– which can get stuck in local optima, to solve discontinuous, non-convex problems. By decomposing the problem using the discontinuities in the source code, Pylic is able to leverage numerical search routines locally to solve the global problem.

The presence of source code allowed us to easily describe meaningful events to the system. For example, in the Marble Maze experiment, to describe the meaningful event of colliding with some specific obstacle we simply provide the system with a predicate that states that the corresponding control-flow statement condition must be true. Without the use of source code, writing a planner would require reimplementation of the non-trivial logic used by the control-flow statement.

## 5 RELATED WORK

Alongside work discussed in the introduction, program structure has been studied in SMT-based approaches, which exploit the structure of the model to perform global search (Inala et al., 2018; Kong et al., 2018; Gao et al., 2013; Shoukry et al., 2017); however, these approaches do not readily leverage domain knowledge. Another approach to exploiting structure is to sample control-flow paths to construct "safety losses" to optimize policy parameters in programs with neural networks and human written code (Yang & Chaudhuri, 2022). Our approach shares similarities with fuzz testing, which has been used to test robotics software (Delgado et al., 2021).

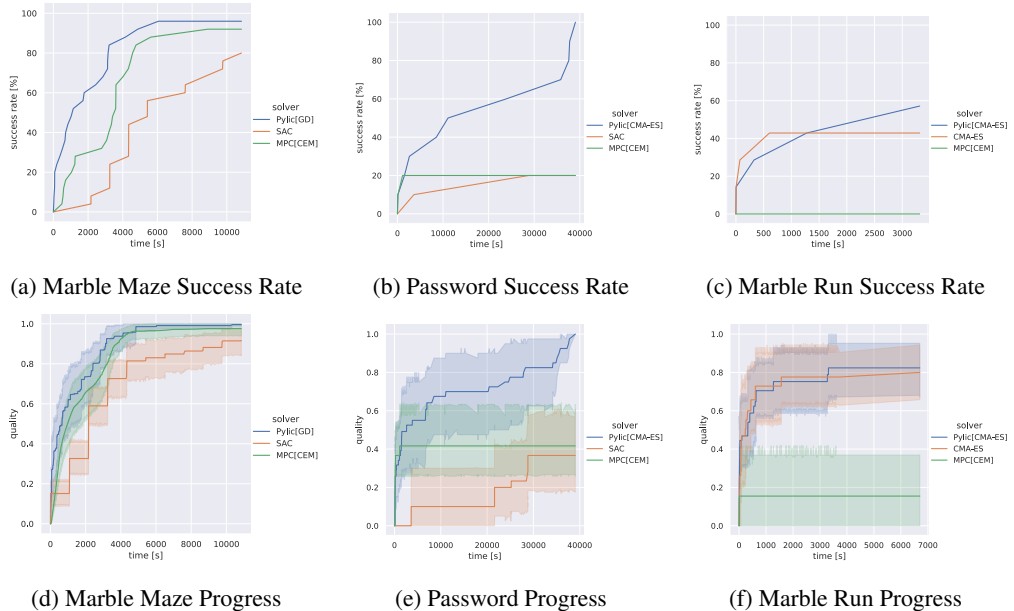

(a) Marble Maze Success Rate      (b) Password Success Rate      (c) Marble Run Success Rate

(d) Marble Maze Progress      (e) Password Progress      (f) Marble Run Progress

Figure 5: Success rates are and progress for all environments. Success rate is the cumulative count of tasks solved. Progress is the median across tasks, with 95% confidence intervals shaded. Note that the success rate is simply the (cumulative) count of instances solved within some time budget.

Previous work has used Linear Temporal Logic formulas on Markov Decision Processes to construct Büchi automata (Cai et al., 2021). Logical specifications have also been studied in compositional RL (Jothimurugan et al., 2021), and to encode domain knowledge (Xie et al., 2021; Li et al., 2021b) and describe constraints in differentiable circuits and neural networks (Ahmed et al., 2022).

Relevant work on Hierarchical Planning includes the abstraction of states to condense irrelevant details in MDPs (Botvinick, 2012; Li et al., 2006; Nashed et al., 2021), or more generally to discover state abstractions (Curtis et al., 2022; Chitnis et al., 2022; Silver et al., 2021), including the use of search trees (Larsson et al., 2020), and multi-scale perception (Hauer et al., 2015). There has also been work that considers planning over parametric primitives with a neural planner and control using a neural trajectory generator (Zhu et al., 2021).

Non-convexity and the presence of discontinuities is a core challenge in robotics (Posa et al., 2014; Wu et al., 2020; Cheng et al., 2022; Marcucci et al., 2017; Aceituno-Cabezas & Rodriguez, 2020; Hogan & Rodriguez, 2020). Recent work has studied the framing of the motion planning problem around obstacles with Convex Optimization (Marcucci et al., 2022). The use of contact modes to guide the search in a sampling-based planning framework has been proposed as an alternative to motion primitives (Cheng et al., 2021), as well as for contact-aware Model Predictive Control (Cleac'h et al., 2021), and tree search with trajectory optimization (Chen et al., 2021). Exploiting discontinuities, such as those found in environments with collisions, has also been studied for collision-resilient multi-copter motion planning (Zha & Mueller, 2021).

## 6 CONCLUSION

We described a framework where the combined planning and control problem is stated directly as a property of the execution of simulator source code. We showed that, despite this generality, it is possible to leverage domain knowledge by relating it to the simulator source code. This allows a tree-based planner to be generated for a given task. Our approach resulted overall in a greater success rate than the numerical search and RL baselines across all three simulated environments. Our method relies on an expert user that can label source code and indicate how the program structure will be used during planning.

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

$$
\begin{array}{lll}
\mathcal{T} & ::= & \mathcal{N} \mid \mathcal{NT} \\
\mathcal{N} & ::= & \textit{IfNode(id, prog\_state, truth\_value)} \\
& \mid & \textit{ForIterStartNode(id, prog\_state, for\_var)} \\
& \mid & \textit{ForIterEndNode(id, prog\_state, for\_var)} \\
& \mid & \textit{ReturnNode(id, prog\_state)} \\
\textit{id} & ::= & \text{node ID as a string} \\
\textit{prog\_state} & ::= & \textit{(var\_name, var\_value) prog\_state} \mid \epsilon \\
\textit{var\_name} & ::= & \text{variable name} \\
\textit{var\_value} & ::= & \text{Python object} \\
\textit{truth\_value} & ::= & \text{floating-point robustness value} \\
\textit{for\_var} & ::= & \text{variable used to iterate}
\end{array}
$$

Grammar 1: Trace grammar.

## A  EXECUTION TRACES

We define a trace to be a sequence of records taken when the control-flow statements of a given program are executed. We consider three types of records: if statements, for loops and return statements. Records contain the program state at recording time and information used to identify the node relative to the program structure. Every record includes (1) an *ID*: a unique identifier of the corresponding control flow instruction, either given by the user or computed from its position in the source code, and (2) *prog_state*: the list of program variables in scope and their values when the statement was executed. Additionally, records that represent a for loop contain the variable used to perform the iteration, and records that represent an if statement contain the truth value of the control condition. Thus, a trace is a sequence as described by Grammar 1.

## B    SEMANTICS

We adapt STL quantitative semantics (Takano et al., 2021; Leung et al., 2021) to our predicate grammar. Given a trace $\tau$, we define the semantics for Table 1 as follows:

$$\rho_B(\neg\phi, \tau) = -\rho_B(\phi, \tau)$$
$$\rho_B(\phi \wedge \psi, \tau) = \min(\rho_B(\phi, \tau), \rho_B(\psi, \tau))$$
$$\rho_B(\phi \vee \psi, \tau) = \max(\rho_B(\phi, \tau), \rho_B(\psi, \tau))$$
$$\rho_B(x < y, \tau) = \rho_{\mathbb{R}}(y, \tau) - \rho_{\mathbb{R}}(x, \tau)$$
$$\rho_B(x > y, \tau) = \rho_{\mathbb{R}}(x, \tau) - \rho_{\mathbb{R}}(y, \tau)$$
$$\rho_B(\text{if\_or}(e), \tau) = \max([n.\textit{truth\_value} \mid n \in \rho_T(e, \tau) \text{ and } n \text{ is } \textit{IfNode}])$$
$$\rho_T(\text{filter}(e, F), \tau) = [n_i \mid n_i \in \rho_T(e, \tau) \text{ and } F(\rho_T(e, \tau), i)]$$
$$\rho_T(\textit{input}, \tau) = \textit{input}$$
$$\rho_{\mathbb{R}}(\mu(e), \tau) = \mu(\rho_T(e, \tau))$$
$$\rho_{\mathbb{R}}(x + y, \tau) = \rho_{\mathbb{R}}(x, \tau) + \rho_{\mathbb{R}}(y, \tau)$$
$$\rho_{\mathbb{R}}(x - y, \tau) = \rho_{\mathbb{R}}(x, \tau) - \rho_{\mathbb{R}}(y, \tau)$$
$$\rho_{\mathbb{R}}(x \times y, \tau) = \rho_{\mathbb{R}}(x, \tau) \times \rho_{\mathbb{R}}(y, \tau)$$
$$\rho_{\mathbb{R}}(x/y, \tau) = \rho_{\mathbb{R}}(x, \tau)/\rho_{\mathbb{R}}(y, \tau)$$
$$\rho_{\mathbb{R}}(c, \tau) = c,$$

where $c \in R$. Note that, for conciseness, in the main text we write $\rho$ instead of $\rho_B$.

$$\frac{\text{def f}(args) \; \{ \; stmt \; \}}{\text{def f}(args, \text{tape}) \; \{ \; stmt \; \}} \; \text{FuncDef}$$

$$\frac{\text{for } (var \text{ in } obj) \; \{ \; stmt \; \}}{\begin{array}{c} \text{for } (var \text{ in } obj) \; \{ \\ \text{record\_for\_begin}(var, \dots) \; ; \; stmt \; ; \\ \text{record\_for\_end}(var, \dots) \; ; \; \} \end{array}} \; \text{ForLoop}$$

$$\frac{\text{if } (b) \; \{ \; stmt \; \}}{\begin{array}{c} nb = \text{Rbst}(b) \; ; \; \text{record\_if}(nb, \dots) \; ; \\ \text{if } (nb > 0) \; \{ \; stmt \; \} \end{array}} \; \text{IfThen}$$

$$\frac{\text{return } e}{v = e \; ; \; \text{record\_return}(v, \dots) \; ; \; \text{return } v} \; \text{Return}$$

(a) Transformation rules for tape manipulation.

$$\frac{a \text{ and } b}{\min(a, b)} \; \text{RbstAnd}$$

$$\frac{a \text{ or } b}{\max(a, b)} \; \text{RbstOr}$$

$$\frac{\text{not } a}{-a} \; \text{RbstNot}$$

$$\frac{a < b}{b - a} \; \text{RbstLt}$$

(b) Transformation rules for boolean expressions.

Figure 6: Transformation rules for execution tracing. Rbst (from "RoBuSTness value") means to apply the matching boolean transformation rule.

## C  PROGRAM TRANSFORMATION RULES

We now describe the operation of the $\text{tr}$ function, which extracts the execution trace of a program under a given input. Our approach is to mechanically instrument the input program by adding code which records the trace into a "tape", which can then be used to evaluate a given predicate.

The required instrumentation follows from these observations: (1) a new variable which points to the tape must be introduced, (2) tracing code which records the program state needs to be added to every control flow expression considered by our system and (3) the expression which computes the truth value of if-statements must be replaced with an equivalent comparison which follows quantitative semantics. These observations result in two groups of program transformations (see Figure 6). The result is a new program with the same semantics which can be executed with an empty tape that gets filled with records as the program is executed.

In our implementation, we use Python's introspection capabilities to read the source code of the input simulation, which is then transformed according to the transformation rules so that it can be traced. This process is automated.

For an example transformation of an if-statement, compare Listing 3 with Listing 4.

Listing 3: Original code

```python
def relu(x):
  if x < 0:  # ID: col
    return 0
  return x
```

Listing 4: Transformed code (simplified)

```python
def relu_transformed(x, tape):
  _rval = pylic.less_than(x, 0)
  tape.append(IfNode(id='col', val=_rval))
  if _rval > 0.0:
    # more tracing code...
    return 0
  # more tracing code...
  return x
```

