# OpenReview forum: "Pylic: Leveraging Source Code for Planning in Structured Environments"
_ICLR.cc/2024/Conference — Submitted to ICLR 2024_

### Official Review · Reviewer_uEX5 · 2023-10-31

**Soundness:** 1 poor
**Presentation:** 2 fair
**Contribution:** 1 poor
**Rating:** 3
**Confidence:** 4

**Summary:**

The paper presents an approach that uses the code of the simulator (or I would the analytical model) and a set of interesting events to solve problems with complex dynamics system. The approach first searches over a sequence of interesting events that reach a goal and then reduces achieving each interesting event (as I understand) as an optimization problem.

**Strengths:**

- The paper is easy to read. It nicely uses a running example to ground the concepts discussed in the paper.

- The paper makes its assumptions clear which makes the paper really understandable.

**Weaknesses:**

While the paper presents attempts to solve an interesting approach, it has a few significant limitations:

- In my opinion, the paper seriously lacks novelty. It invents (or introduces) new terms for concepts that have long existed. E.g., the main contribution claimed by the paper is using **code of the simulator** for solving the problem faster. However, this is nothing but having access to an analytical model of the system. Why to complicate the paper? The second term would be the **meaningful events**. The events that are required or necessary to be achieved in order to reach the goal state. These is analogous to landmarks [1] or critical regions [2]. Landmarks and critical regions have been extensively used in planning and robotics literature.

- The approach requires the analytical of the model of the system as well as a set of landmarks or critical regions to be provided upfront. This does not only require a domain expert at the train time but at the test time as well. Which is infeasible to have. Especially, when a lot of research has been focused on learning these landmarks or critical regions automatically as well as approaches than learn policy without explicitly having access to an analytical model of the environment and treating the simulator as a blackbox.

- It is not clear from the paper that how a sequence of low-level action is generated to reach each meaningful event. My educated guess is the problem is reduced to an optimization problem but it has to be clear from the paper.

- Lastly, the empirical evaluation is extremely weak. Especially, the choice of the baselines. Given that this approach is a model-based optimization approach. This should be compared with a hierarchical planning approach [2,3,4] or a hierarchical optimization approach [5] or a model-based RL approach.

### References


[1] Hoffmann, Jörg, Julie Porteous, and Laura Sebastia. "Ordered landmarks in planning." Journal of Artificial Intelligence Research 22 (2004): 215-278.

[2] Shah, Naman, and Siddharth Srivastava. "Using Deep Learning to Bootstrap Abstractions for Hierarchical Robot Planning." Proceedings of the 21st International Conference on Autonomous Agents and Multiagent Systems. 2022.

[3] Garrett, Caelan Reed, Tomás Lozano-Pérez, and Leslie Pack Kaelbling. "Pddlstream: Integrating symbolic planners and blackbox samplers via optimistic adaptive planning." Proceedings of the International Conference on Automated Planning and Scheduling. Vol. 30. 2020.

[4] Shah, Naman, et al. "Anytime integrated task and motion policies for stochastic environments." 2020 IEEE International Conference on Robotics and Automation (ICRA). IEEE, 2020.

[5] Toussaint, Marc. "Logic-Geometric Programming: An Optimization-Based Approach to Combined Task and Motion Planning." IJCAI. 2015.

**Questions:**

Please refer to the previous section.

---

> ### Author Response · Authors · 2023-11-20
>
> We greatly appreciate your insightful comments and questions, which have helped us improve our manuscript and also inform ongoing efforts. We follow up on your comments below.
>
> ### On simulator code
>
> We agree that source code is ultimately an analytical model. However, we argue that source code has key features that make it unlike other forms of analytical models. Specifically, our focus on leveraging control flow makes it possible to easily reuse parts of the model when communicating domain knowledge. Our claim is that this allows complex search spaces to be described efficiently by an expert user, allowing the system to find solutions to a problem domain, as shown in the experiments.
>
> This is why the focus on source code is not inconsequential and is not equivalent to a general analytical model with no assumptions about its internal structure.
>
> ### On meaningful events
>
> We appreciate the connection to landmarks and critical regions, and we agree that the concept of meaningful events is related to concepts used in other planning frameworks. Note that our "meaningful events", however, have additional assumptions specific to our framework, which would make the use of, e.g., the terms "landmarks" or "critical regions" incorrect (even if, broadly speaking, they are related). This is why we decided to use a different term. We have revised our manuscript to explicitly reference the related concepts.
>
> ### On requiring a domain expert at test time
> (Also included in response to Reviewer ppCk)
>
> We want to clarify that the domain knowledge provided to the system is general to a problem domain and not bound to a particular problem instance. We believe this important point was not efficiently communicated in our manuscript. Specifically, the user provides general principles to search over sequences of meaningful events for an entire problem domain (section 3.2). We have revised our manuscript to clarify this.
>
> Thus, we do not "require" the domain expert for any particular task; the expert is only required to describe the search over sequences of meaningful events for a problem domain.
>
> ### On generating the sequence of low-level actions
>
> Your assumption is correct; the problem is reduced to an optimization problem. This is described in Section 3.1 (now 3.2). We have revised the manuscript to make the description of the framework easier to follow.
>
> ### On the choice of baselines
> (Also included in response to Reviewer wE34)
>
> We agree that comparing our approach with pure RL and MPC approaches is not completely straightforward because we allow users to provide structured domain knowledge. We also agree that the techniques you list would be good candidates for further benchmarking.
>
> Specifically, we are working on a comparison of our technique with planning techniques that leverage temporal specifications. This is more of an apples-to-apples comparison because some of these techniques also allow (and rely) on a human to provide the domain knowledge to make a domain of tasks feasible, and this knowledge is encoded at a comparable level of abstraction.

---

> > ### Comment · Reviewer_uEX5 · 2023-11-22
> > **Response**
> >
> > I appreciate authors' response to the issues raised. The response has resolved some of the issues, however, I still do not agree with authors' on some of the points.
> >
> > - Authors still claim that "meaningful events" and "landmarks" or "critical regions" are different but they do not provide any information on how.
> >
> > - The authors claims that the approach does not require an expert at the test time but I do not agree. Meaningful events are different for different problems in the same domain. The alternative is to exhaustively list all possible meaningful events a priori which I don't feel is possible neither mentioned in the paper.
> >
> > - The authors claim that their reason for using simulator code and not the analytical model ( IMO both are the same) is to encourage reuse. However, this is not at all clear from the paper. Here the expert just not have to be expert in the domain but also at the simulator. There details that are important and in a way significant to your approach needs to be extremely clear from the paper.

---

> > > ### Author Response · Authors · 2023-11-22
> > >
> > > We express our gratitude to the reviewer for the follow-up response, and the opportunity to better understand and address the issues raised.
> > >
> > > ## On expert at test time
> > >
> > > You are correct in that, for different problems in the same domain, different sequences of meaningful events are needed to solve any particular task. However, note that the expert does not provide a particular sequence of meaningful events or even a fixed set of meaningful events. Rather, the expert describes how to construct sequences of meaningful events incrementally  ("GROW" routine in Algorithm 1). In this way, the system can search for sequences of meaningful events on its own.
> > >
> > > In other words, the expert does not have to anticipate what a solution to any specific task would look like *exactly*; instead, it has to anticipate the *structure* of (possibly) useful sequences of meaningful events to incrementally search the solution space. Effectively, the expert is tasked with providing a single program ("GROW") for the entire problem domain that receives a candidate partial solution to any problem instance and answers the question, "Given *this* partial solution, what are some meaningful events the system could try solving for next?", which the system can then use to search over sequences by performing a tree search (a standard approach).
> > >
> > > This avoids the problem of listing all possible meaningful events, which we agree is not feasible.
> > >
> > > ## On "meaningful events", "landmarks," and "critical regions"
> > >
> > > Our motivation for introducing the term "meaningful events" is two-fold:
> > >
> > > - Central to our definition is the constraint that a "meaningful event" is something that a domain expert can describe as a "precise statement about the execution of the simulator[...]". With this definition, we intend to highlight two core assumptions of our framework: there is an expert user capable of relating domain knowledge to source code, and these statements relate to the execution of a structured model (simulator source code). Introducing the term "meaningful events" allows us to highlight these two non-trivial assumptions.
> > >
> > > - Furthermore, we agree that "meaningful events" could be understood --e.g.,-- as a special case of "landmarks" with a "reasonable order". The key observation is that we are interested in describing properties about the execution of the model, not only about the simulated system.  We are then in (what would be) the special case where the set of atomic facts consists of statements about the execution of the structured model (further, note that this set is not finite) and the actions consist of a change over the control parameters. Thus, describing our approach from within the landmark framework would make the explanation unnecessarily verbose and non-intuitive, and introducing "meaningful events" as a self-contained concept leads to a more succinct description of our approach (note that we nonetheless updated the manuscript to acknowledge and cite these analogous concepts).
> > >
> > > Because of these reasons, we argue, it is appropriate to introduce the term "meaningful events" for the purpose of describing the framework.
> > >
> > > ## On highlighting code reuse
> > >
> > > We have revised the manuscript to make instances of code reuse explicit, and appreciate the observation that this was not sufficiently emphasized in the previous version.
> > >
> > > These are the key instances of code reuse in the paper (and corresponding changes to the manuscript):
> > >
> > > - Motivating example section: we describe how the problem of "finding control parameters that collide with an obstacle" corresponds to "finding an input to the simulator such that the collision control flow is true". This is a good example of code reuse because the alternative is to reimplement the non-trivial collision logic. Instead, our approach allows the user to leverage the existing logic when describing sequences of meaningful events. We have updated our manuscript to highlight this example.
> > > - Methods section: the design of the trace predicate grammar includes constructs to index into the structure of the source code, precisely to allow reuse. We updated the manuscript to highlight this key motivation of the grammar's design.
> > > - Experiments section: in each experiment, the GROW routine leverages the source code structure to build the trace predicates; however, this was not explicit. We've updated the manuscript, adding a description of how code was reused when describing the trace predicates.
> > >
> > > Finally, we agree that the assumption that the domain expert can understand the simulator code is non-trivial. Without disputing this fact, we note that (1) it is common for robotics practitioners to be proficient programmers, (2) some existing planning frameworks are also non-trivial to use for non-programmers (e.g., PDDL approaches), and (3) we only require users to understand the piece of code that is relevant for a given problem domain (as shown in the experiments where the low-level rigid body dynamics are treated as a black box).

---

### Official Review · Reviewer_ppCk · 2023-11-01

**Soundness:** 2 fair
**Presentation:** 2 fair
**Contribution:** 2 fair
**Rating:** 3
**Confidence:** 4

**Summary:**

The paper describes an approach that utilizes code inspection techniques to locate discontinuities in a task together with user-provided critical junction points to formulate a tree-based search problem. The approach assumes that solutions between junction points can be found by local numerical search methods while the global sequencing is guided by the user-provided "meaningful events".

**Strengths:**

The general idea of the work is interesting in that it attempts to leverage program verification approaches and logic to solve complex global optimization problems.

**Weaknesses:**

While the general idea is interesting, there are many issues with the paper in its current form.

The paper argues that other methods rely on specialized representations, which makes them hard or inconvenient to use. However, the proposed method requires the user to specify so-called "meaningful events". Judging from the examples and the description, this would appear to be an even more onerous requirement as they have to be defined by the end-user instead of the designer of a general method for particular problem scenarios. This aspect is insufficiently discussed in the paper, making it unclear that this approach is practical in contrast to other methods such as reinforcement learning, model-based control, or Monte Carlo tree search.

The core aspect of the work revolves around tracing the code logic to find control flow statements. In that context, the paper repeatedly mentions simulators as the code to be traced. However, in the experiments, only one example traces through something akin to a simulator and in all other instances, some generic logic code is traced. Therefore, the mention of simulators is quite confusing as in no instance is a proper simulator, such as pybullet, mujoco, Isaac Sim, drake, etc., traced. It also remains unclear that the discontinuities in the simulation that this process should find are necessarily "telegraphed" by control flow statements rather than pure linear algebra, which the proposed approach would not appear to register. Tracing general program execution can still be interesting, as the experiments show. However, the description of the applications and properties described in the main body of the text is misleading. Without seeing the experiments, I would have expected the proposed method to be able to trace through complex physics engines as employed by Isaac sim or pybullet.

The proposed method uses quite a few components and joins them together. While an overview is provided in Figure 1, this figure is never used in the text to help the reader understand how things connect. As such, it is hard to follow where the different pieces go and how they interact. For example, there is a connection between user-defined events, code tracing, and trace predicates. This can be gleaned from the text to some extent, but making the connections more easily understood and more evident would improve the readability of the paper significantly.

The paper states that the local search is sufficient to find parameters to reach the next meaningful event. However, it is not mentioned how this can be guaranteed or why this should hold in the first place. Are there theoretical guarantees that can alert the user when this is impossible, or does the user have to add "meaningful event" specifications until things can be solved?

The experimental section, while containing several experiments, is lacking in detail. There are detailed descriptions of the experimental setups, yet the discussion of the results is unsatisfactory as they provide no real insight. Furthermore, the choice of baselines and problem setups is perplexing. The biggest issue is that some of the experiments would be ideally suited for Monte Carlo tree search methods, especially given the tree search nature of the proposed system, yet approaches based on this technique are absent. Another aspect is that the problem setups for different methods are not identical,  making it unclear whether the results are comparable. A good example of this is 4.2, where the proposed method operates on a state representation of button states while the RL and MPC baseline operate on an entirely different state space.

While the idea, in general, is interesting, I cannot recommend this paper for publication in its current state.

**Questions:**

- Some of the description and experimental tasks used give a task and motion planning vibe, would such tasks and methods be sensible comparisons for this work?
- Is the need to have traced control flow labels, predicates, and user-specified "meaningful events" not more challenging and domain-specific than representations required by other approaches?
- How can the assumption of local searches finding connections between the sequence of "meaningful events" be guaranteed?

---

> ### Author Response · Authors · 2023-11-20
>
> We greatly appreciate your insightful comments and questions, which have helped us improve our manuscript and also inform ongoing efforts. We follow up on your comments below.
>
> ### Question 1: baselines
> (Also included in response to Reviewer wE34)
>
> As you correctly point out, our proposed approach currently relies on a human to provide the space of meaningful events for a given problem domain (and relate them to the source code). Thus, comparing our approach with pure RL and MPC approaches is not completely straightforward because we allow users to provide structured domain knowledge.
>
> We are working on a comparison with planning techniques that leverage temporal specifications. This is more of an apples-to-apples comparison because some of these techniques also allow (and rely) on a human to provide the domain knowledge to make a domain of tasks feasible, and this knowledge is encoded at a comparable level of abstraction.
>
> ### Question 2: user effort
> (Also included in response to Reviewer wE34)
>
> We argue that the amount of effort required by our approach is similar to, or smaller than, techniques that leverage domain knowledge during planning. This is because, in our framework, the user can encode complex conditions (e.g., a collision) simply by referencing the corresponding constructs in the meaningful event predicates without reimplementing them from scratch (provided that such constructs are present in the simulator, which is one of our assumptions).
>
> ### Question 3: How can the assumption of local searches finding connections between the sequence of "meaningful events" be guaranteed?
>
> We assume that the user has provided events for which this assumption is true. We have revised our manuscript to further highlight this assumption.
>
> In the special case where meaningful events induce convex problems, the connections between events can be guaranteed to be reachable with gradient descent. However, in general, the user is subject to the limitations of the optimization method used to define meaningful events.
>
> ### On specifying meaningful events
>
> We want to clarify that the domain knowledge provided to the system is general to a problem domain and not bound to a particular problem instance. We believe this important point was not efficiently communicated in our manuscript. Specifically, the user provides general principles to search over sequences of meaningful events for an entire problem domain (section 3.2). We have revised our manuscript to clarify this.
>
> ### On tracing simulators
>
> It is precisely the logic that reflects the structure of a problem domain that can be related to domain knowledge. As we show, this key logic can be low-level dynamics (e.g., collision handling), but more often it will be logic that leverages a low-level physics engine (e.g., the button dynamics in "Password locomotion"). This is why, in the experiments, we explain how we leverage code at the appropriate level of abstraction for a given problem domain.
>
> ### On the connections between components of the system
>
> The different components come together in Algorithm 2 (now Algorithm 1). We have revised the manuscript Methods section to make the description easier to follow and the connections between components more explicit.

---

### Official Review · Reviewer_wE34 · 2023-11-01

**Soundness:** 2 fair
**Presentation:** 2 fair
**Contribution:** 2 fair
**Rating:** 5
**Confidence:** 2

**Summary:**

The paper presents a novel method of using trace information from the source code of a simulator to more efficiently solve planning problems. The method is evaluated on three simulation examples where it compares favorably to RL (SAC) and sampling-based MPC.

**Strengths:**

- The idea of using the source code of a simulator to speed up planning is interesting and appears novel.
- The paper is fairly well-written, but there are a lot of things going on
- The benchmark results are encouraging

**Weaknesses:**

The fundamental limitation of this approach is that it needs a human to extract what they call "meaningful events" that serves as the - foundation of the planning tree. This makes comparisons against model-free methods like RL and sampling-based MPC rather apples-to-oranges. If this step was more automatic, or shown to be very simple, I think the paper would be much stronger.
- This is exacerbated by not using any standard benchmarks that I can see.
- The performance difference compared to model-free approaches also does not appear that large in two of three experiments. Ultimately if this is useful or not probably depends on the users proficiency in the syntax of the proposed framework and the level of understanding of the simulator code. Not sure if a user study might help.

Minor:
- The success rate curves could also use a confidence interval (or quartiles).

**Questions:**

- Why did you not include any standard benchmark environments from e.g. RL as you are comparing against RL?

---

> ### Author Response · Authors · 2023-11-20
>
> We greatly appreciate your insightful comments and questions, which have helped us improve our manuscript and also inform ongoing efforts. We follow up on your comments below.
>
> ### On "comparison with RL and MPC is not apples-to-apples"
>
> As you correctly point out, our proposed approach currently relies on a human to provide the space of meaningful events for a given problem domain (and relate them to the source code). Thus, comparing our approach with pure RL and MPC approaches is not completely straightforward because we allow users to provide structured domain knowledge.
>
> We are working on a comparison with planning techniques that leverage temporal specifications. This is more of an apples-to-apples comparison because some of these techniques also allow (and rely) on a human to provide the domain knowledge to make a domain of tasks feasible, and this knowledge is encoded at a comparable level of abstraction.
>
> Finally, we argue that the amount of effort required by our approach is similar to, or smaller than, that required by these techniques. This is because the user can encode complex conditions (e.g., a collision) simply by referencing the corresponding constructs in the meaningful event predicates without reimplementing them from scratch (provided that such constructs are present in the simulator, which is one of our assumptions).
>
> ### On "standard RL benchmarks"
>
> The reason we did not include the standard RL benchmarks is that our tool requires the code that has to be traced to be Python code. However, in the OpenAI Gym continuous tasks (e.g., half-cheetah, hopper, etc.), the code that encodes the structure of the task is the Mujoco simulator, which is a binary of C source code.
>
> Because of this, we designed the --arguably harder-- "Password Locomotion" task, which leverages the rigid body dynamics as part of a more complex simulation. The simulator makes use of Mujoco but encodes the logic that handles the dynamics of pressing buttons and unlocking the goal in Python.
>
> ### On success rates curves confidence intervals
>
> These curves are an exact measurement: they are the cumulative number of tasks that are solved over the course of the experiment (which itself consists of multiple tasks). In other words, these curves correspond exactly to the count of successful tasks as the experiment progresses (i.e., they are not an aggregate value), and thus have no confidence interval associated.

---

> > ### Comment · Reviewer_wE34 · 2023-11-20
> > **Response to authors**
> >
> > I thank the authors for the candid responses. My score remains unchanged as I think the paper needs to do more to show practical utility. The other reviewers also had some good suggestions. One idea could be a user study where you show that this is less work, however not comparing on any standard benchmark because you require python is also a pretty big problem. Maybe you can find at least some benchmarks from other papers that are only in Python. E.g. the simple ones like Lander, pendulum swing-up etc. I'm sure there are more.

---

### Meta-Review · Area_Chair_9TQr · 2023-12-09

**Metareview:**

This is a very interesting paper that leverages access to the source code of a physics simulator, and in particular, access to the control flow and execution traces of programs, given an input sequence (actions/controls and any physics parameters). "Meaningful events" such as contacts or other key events can be either automatically detected in the program or manually annotated by users, on whose annotations the proposed method does currently rely. Controls are optimized via gradient descent, guided towards satisfying constraints imposed by intermediate key events, according to an automatically generated plan over the meaningful events of the simulator (performed via tree search). Despite the very interesting premise of paper, there are a few major issues that have to be addressed:


1. The paper does not compare with existing Task and Motion Planning approaches, such as Logic Geometric Programming or PDDLStream solvers. In fact, the writing of the paper is not very much aligned with existing literature on task and motion planning or planning through contacts.


2. Despite the rebuttal discussion, it is not quite clear how much effort users have to spend to annotate key events. Is it more work to specify logic rules than to annotate these events? It would seem so, but the paper needs to do a better job to show this in a reliable way.


3. Despite the improvements in writing, and the pedagogical example, the writing of the paper can be improved significantly. For example, how is backtracking handled in the tree search? Algorithm 1 keeps a set of failed events, but it is not clear why a tree node is disqualified if only one attempt at continuous motion optimization fails (what if the optimization of controls hits a local minimum but the event could be realized if a better local optimum was found)?

4. Reviewers expressed concerns about the reliance on user annotations, but I think it is a surmountable problem in future iterations of this work.

So, overall, this is a very interesting paper, but it needs one more iteration of work before it is useful to an robotics and planning audience.

**Justification For Why Not Higher Score:**

See reasons 1-4 above.

**Justification For Why Not Lower Score:**

N/A

---

### Decision · Program_Chairs · 2024-01-16

Reject